# Elicitation of Cluster A and Co-Receptor Binding Site Antibodies Are Required to Eliminate HIV-1 Infected Cells

**DOI:** 10.3390/microorganisms8050710

**Published:** 2020-05-11

**Authors:** Guillaume Beaudoin-Bussières, Jérémie Prévost, Gabrielle Gendron-Lepage, Bruno Melillo, Junhua Chen, Amos B. Smith III, Marzena Pazgier, Andrés Finzi

**Affiliations:** 1Centre de Recherche du CHUM, Montreal, QC H2X 0A9, Canada; guillaume.beaudoin-bussieres@umontreal.ca (G.B.-B.); jeremie.prevost@umontreal.ca (J.P.); gaby_gendron@yahoo.ca (G.G.-L.); 2Département de Microbiologie, Infectiologie et Immunologie, Université de Montréal, Montreal, QC H3T 1J4, Canada; 3Department of Chemistry, School of Arts and Sciences, University of Pennsylvania, Philadelphia, PA 19104-6323, USA; bmelillo@sas.upenn.edu (B.M.); jhchen@sas.upenn.edu (J.C.); smithab@sas.upenn.edu (A.B.S.III); 4Infectious Diseases Division, Uniformed Services University of the Health Sciences, Bethesda, MD 20814, USA; marzena.pazgier@usuhs.edu

**Keywords:** HIV-1, ADCC, cluster A, coreceptor binding site, guinea pigs, small CD4 mimetics

## Abstract

HIV-1-infected individuals raise a polyclonal antibody response targeting multiple envelope glycoprotein (Env) epitopes. Interestingly, two classes of non-neutralizing CD4-induced (CD4i) antibodies, present in the majority of HIV-1-infected individuals have been described to mediate antibody-dependent cellular cytotoxicity (ADCC) in the presence of small CD4 mimetic compounds (CD4mc). These antibodies recognize the coreceptor binding site (CoRBS) and the constant region one and two (C1C2 or inner domain cluster A) of the gp120. In combination with CD4mc they have been shown to stabilize an antibody-vulnerable Env conformation, known as State 2A. Here we evaluated the importance of these two families of Abs in ADCC responses by immunizing guinea pigs with gp120 immunogens that have been modified to elicit or not these types of antibodies. Underlying the importance of anti-CoRBS and anti-cluster A Abs in stabilizing State 2A, ADCC responses were only observed in the presence of these two types of CD4i antibodies. Altogether, our results suggest that these two families of CD4i antibodies must be taken into account when considering future strategies relying on the use of CD4mc to eliminate HIV-1-infected cells in vivo.

## 1. Introduction

The envelope glycoproteins (Env) of the human immunodeficiency virus type 1 (HIV-1) mediate viral fusion and are assembled into a trimer of heterodimers composed of the transmembrane gp41 and exterior gp120 subunits [1,2]. Env is the only virus-specific antigen at the surface of viral particles and HIV-1 infected cells [3], and therefore is the target of neutralizing and non-neutralizing antibodies. Current vaccine approaches aim to elicit broadly neutralizing antibodies (bNAbs) against Env [4,5]. So far, no vaccine strategy succeeded in eliciting such antibodies in humans [5]. The only anti-HIV-1 vaccine trial that presented a modest level of protection (31.2%), the RV144 trial [6], elicited non-neutralizing antibodies (nnAbs). In this trial, evidence suggested that the induction of antibodies with enhanced Fcγ receptor (FcγR)-mediated activities might have contributed to the protection [7,8,9,10]. Indeed, correlative evidence suggests that antibodies to V1V2 and CD4-induced (CD4i) epitopes may have contributed to protection [8,9]. Although antibody-dependent cellular cytotoxicity (ADCC) was not identified as a correlate of protection in the RV144 trial, a non-significant trend towards a lower risk of HIV-1 infection was observed among vaccine recipients with higher ADCC responses and low IgA responses [9]. Whether this was associated with the presence of a histidine at position 375 (H375) of the circulating CRF01_AE strain in Thailand, which predisposes Env to spontaneously assume a more “open” and ADCC-susceptible conformation remains unclear [11]. 

The difficulty in targeting HIV-1 by vaccination is many folds: HIV-1 has a high rate of mutations resulting in the evasion of the humoral response of the host [4]. The high levels of Env glycosylation also help the virus evade the antibody response by rendering the binding of the antibodies more difficult and by mimicking the host self [12,13,14]. Env conformational masking helps protect the conserved regions from antibody binding [15]. Env flexibility also contributes to antibody evasion [15]. Indeed, Env is a metastable molecular complex that transitions from the unliganded “closed” high-energy conformation (state 1) to an “open” CD4-bound low-energy conformation (state 3). CD4 engagement drives Env into an intermediate “partially open” conformation (state 2) and then into state 3 [16,17]. The unliganded form of Env from primary HIV-1 isolates assumes a “closed” state 1 conformation that is mostly resistant to antibody attack [17], but can be recognized by broadly neutralizing antibodies. CD4 binding “opens” Env by reorganizing the V1V2 and V3 loops resulting in the adoption of the CD4-bound conformation, referred as state 3 [16,17,18]. This conformation is vulnerable to ADCC mediated by CD4-induced (CD4i) antibodies easily elicited by vaccination or present in the sera from most HIV-1-infected individuals [19,20,21,22]. The problem of exploiting this conformation for vaccine purposes remains the difficulty in stabilizing Env in an “open” conformation. A promising approach to achieve this is the use of small molecule CD4 mimetics (CD4mc) which binds into the gp120 Phe43 cavity and induce conformational changes resulting in the exposure of several CD4i epitopes [23,24,25,26]. This strategy was recently used to prove that CD4mc can synergize with antibodies elicited by monomeric gp120 to protect non-human primates (NHPs) from multiple high-dose intrarectal challenges with a heterologous simian-human immunodeficiency virus (SHIV) [21]. In this study, the vaccine-elicited anti-gp120 Abs alone were unable to neutralize viral particles or mediate ADCC, consequently, they failed to protect monkeys from SHIV challenges. Strikingly, the same non-neutralizing gp120-elicited Abs combined with CD4mc protected monkeys from multiple heterologous SHIV challenges [21]. We recently reported that CD4mc in combination with two types of CD4i anti-gp120 antibodies stabilize a new Env conformation, state 2A, that is vulnerable to antibody attack [26,27]. Specifically, this conformation is stabilized by CD4mc in combination with anti-co-receptor binding site (CoRBS) and anti-cluster A antibodies [26,28]. Mechanistic studies have shown that CD4mc initially exposes the CoRBS but not the cluster A region, which is located in the gp120 inner domain. Anti-CoRBS Ab binding further “opens” the trimeric Env exposing the cluster A region, thus stabilizing state 2A [26,29]. 

To further characterize the requirement of these two families of Abs to eliminate infected cells by ADCC in vaccine settings, we immunized guinea pigs with gp120 immunogens having a differential ability to elicit CD4i CoRBS and cluster A Abs. Serum from immunized guinea pigs was then tested for their ability to recognize Env and mediate ADCC in the presence or absence of the potent CD4mc (BNM-III-170) [30]. Our results stress the importance of eliciting both type of antibodies that in combination with CD4mc are able to effectively eliminate HIV-1-infected cells by ADCC.

## 2. Materials and Methods

### 2.1. Plasmids

The codon-optimized pcDNA3.1-HIV-1_YU2_ gp120 ΔV1V2V3V5 (gp120_core_) plasmid was made by replacing the sequence encoding residues 124 to 198 from the V1/V2 loop with a sequence encoding a GG linker, the sequence encoding residues 302 to 323 from the V3 loop with a sequence encoding a GGSGSG linker and the sequence encoding residues 460 to 465 from the V5 loop with a GSG linker [31,32]. The W69A mutation was introduced into pcDNA3.1-HIV-1_YU2_ ΔV1V2V3V5 by site-directed mutagenesis using the QuikChange II XL protocol (Stratagene, La Jolla, CA, USA). The gp120 inner domain constructs ID2 consists of residues 44–118 and 206–256 of gp120 connected by a GGA linker that is connected to the residues 472–496 of gp120 by another GG linker to remove the outer domain. ID2 was further stabilized by introducing a disulfide bond with two mutations (V65C and S115C) [33]. All mutations were confirmed by Sanger DNA sequencing. The pSVIIIenv vector expressing HIV-1_YU2_ envelope glycoproteins with a truncated cytoplasmic tail (ΔCT) and a pcDNA3.1 human CD4 expressor were previously described [34]. The pNL43-ADA(Env)-GFP.IRES.Nef proviral vector and the VSV-G-encoding plasmid (pSVCMV-IN-VSV-G) were previously described [20].

### 2.2. Production of Recombinant gp120

FreeStyle 293F cells (Thermo Fisher Scientific, Waltham, MA, USA) were grown in FreeStyle 293F medium (Thermo Fisher Scientific, Waltham, MA, USA) to a density of 1 × 10^6^ cells/mL at 37 °C with 8 % CO_2_ with regular agitation (150 rpm). Cells were transfected with recombinant gp120 expressors using ExpiFectamine 293 transfection reagent, as directed by the manufacturer (Thermo Fisher Scientific, Waltham, MA, USA). One week later, cells were pelleted and supernatants were filtered using a 0,22-µm-pore-size filter (Thermo Fisher Scientific, Waltham, MA, USA). The gp120 glycoproteins were purified by nickel affinity columns, as directed by the manufacturer (Thermo Fisher Scientific, Waltham, MA, USA). Monomeric gp120 was subsequently purified by fast protein liquid chromatography (FPLC), as previously reported [32]. The purification by FPLC was performed using an AKTA Prime Plus FPLC with a HILOAD 16/60 Superdex 200 PG (General Electric, Boston, MA, USA). The gp120 preparations (called later in the paper gp120_core_) were dialyzed against phosphate-buffered saline (PBS) and stored in aliquots at −80 °C until further use.

### 2.3. Guinea Pig Immunization

The standard 108-Day immunization protocol of Cocalico Biologicals, Inc. (CBI, Reamstown, PA, USA) for guinea pigs was followed. Hartley guinea pigs (~7 weeks of age) were housed at the animal facility of CBI. 18 guinea pigs were assigned evenly into 3 groups: inoculated intramuscularly with 25 µg of gp120_core_, gp120_core_ W69A, or sterile PBS. The immunogens were emulsified with AS02A adjuvant in a final volume of 500 µL. The immunogen-adjuvant emulsion was prepared between 60 and 120 min before inoculation into the animals. Three supplemental immunizations by muscular injections were carried out following the first injection. Blood samples were collected from all guinea pigs on Day 108 as final bleed sera samples following the fourth immunization on Day 98. The immunization protocol is outlined in Figure 1. The sera were isolated and incubated at 55 °C for 60 min to heat-inactivate complement and stored at −20 °C. All animal experiments were performed according to the protocol (Project Number 2018-0788) approved by the IACUC of Cocalico Biologicals Animal Care and Use Committee on 08/28/2018.

### 2.4. ELISA

Bovine serum albumin (BSA), gp120_core_ and stabilized gp120 inner domain (ID2 [33]) were prepared in PBS (0.1 µg/mL) and adsorbed to MaxiSorp; Nunc plates (Thermo Fisher Scientific, Waltham, MA, USA) overnight at 4 °C. BSA was used as a negative control. Coated wells were subsequently blocked with blocking buffer (Tris-buffered saline (TBS) containing 0.1% Tween 20 and 2% [wt/vol] BSA) for 90 min at room temperature. Wells were then washed 4 times with washing buffer (Tris-buffered saline [TBS] containing 0.1% Tween 20). Sera from the final bleed of the immunized guinea pigs (1:10,000 dilution) were diluted in blocking buffer and incubated for 120 min at room temperature. Wells were then washed 4 times with washing buffer. This was followed by incubation of horseradish peroxidase (HRP)-conjugated antibody specific for the Fc region of guinea pig IgG (0.4 µg/mL; Thermo Fisher Scientific, Waltham, MA, USA) for 90 min at room temperature. Wells were then washed 4 times with washing buffer. HRP enzyme activity was determined after the addition of a 1:1 mix of Western Lightning ECL reagents (Perkin Elmer Life Sciences, Waltham, MA, USA). Light emission was measured with an LB 941 TriStar luminometer (Berthold Technologies, Bad Wildbad, Germany). 

### 2.5. Cell Lines and Isolation of Primary CD4+ T Cells

HEK293T human kidney cells (obtained from ATCC, American Type Culture Collection, Manassas, VA, USA) were grown at 37 °C and 5% CO_2_ in Dulbecco′s modified Eagle′s medium (DMEM) (Thermo Fisher Scientific, Waltham, MA, USA) containing 5% fetal bovine serum (FBS) (MilliporeSigma, St. Louis, MO, USA) and 100 µg/mL of penicillin-streptomycin (Wisent, Montreal, QC, CA). PBMCs were obtained from HIV-1-uninfected individuals through leukapheresis. The leukapheresis were performed under regulation provided by CRCHUM and written consent was obtained for each donor. Primary CD4+ T lymphocytes were purified from rested PBMCs by negative selection with EasySep Human CD4+ T Cell Isolation Kit (Stemcell Technologies, Vancouver, BC, CA). Primary CD4+ T lymphocytes were activated with phytohemagglutinin-L (PHA-L) (10 µg/mL) for 48 h to Roswell Park Memorial Institute (RPMI) 1640 Medium (Thermo Fisher Scientific, Waltham, MA, USA) containing 20% fetal bovine serum (FBS) (MilliporeSigma, St. Louis, MO, USA) and 100 µg/mL of penicillin-streptomycin (Wisent, Montreal, QC, CA). Primary CD4 T lymphocytes were then cultured in complete RPMI supplemented with recombinant interleukin-2 (rIL-2) (100U/mL) at 37 °C and 5% CO_2_.

### 2.6. Viral Production and Infection 

Vesicular stomatitis virus G (VSVG)-pseudotyped NL4.3 green fluorescent protein (GFP) ADA-Env-based viruses were produced by transfection in HEK293T cells as described previously [20,34]. Forty-eight hours after transfection, cell supernatants were harvested, clarified by low-speed centrifugation (4 min at 1500 rpm), and concentrated by ultracentrifugation for 60 min at 4 °C at 29,000 rpm on a 20% sucrose cushion. Pellets were harvested in fresh RPMI, and aliquots were stored at −80 °C until use. Viruses were then used to infect approximately 10% of primary CD4 T cells from HIV-1 negative donors by spin infection at 800x g for 60 min in 96-V well plates at 25 °C.

### 2.7. Cell Surface Staining

For cell surface staining of Env-transfected cells, three million HEK293T cells were seeded in 100 mm petri dishes (MilliporeSigma, St. Louis, MO, USA). Twenty-four hours later, cells were transfected by the standard calcium phosphate method. Forty-eight hours post-transfection, HEK293T cells were incubated for 45 min at 37 °C with guinea pig sera (1:500). Cells were then washed twice with PBS and stained with 2 µg/mL of anti-guinea pig AlexaFluor 647 (Thermo Fisher Scientific, Waltham, MA, USA) (AF-647) secondary antibody (which does not cross-react with human Abs) and 1:1000 dilution of viability dye AquaVivid (Thermo Fisher Scientific, Waltham, MA, USA) for 20 min in PBS at room temperature. After two more PBS washes, cells were fixed in a 2% PBS-formaldehyde solution. Alternatively, cells were preincubated with A32 or 17b (5 µg/mL) for 30 min, prior staining with guinea pig sera. For cell surface staining of HIV-1-infected cells, forty-eight hours post-infection, primary CD4+ T cells infected with NL4.3 ADA GFP WT were stained for 30 min at 37 °C with guinea pig sera (1:1000 dilution) in PBS with or without 50 µM of BNM-III-170 and/or 1 µg/mL of 17b. Cells were then washed once with PBS and stained with 2 µg/mL anti-guinea pig AlexaFluor 647 (AF-647) secondary antibodies and 1:1000 dilution of viability dye AquaVivid (Thermo Fisher Scientific, Waltham, MA, USA) for 20 min in PBS at room temperature. Cells were then washed with PBS and fixed in a 2% PBS-formaldehyde solution. All samples were acquired on an LSRII cytometer (BD Biosciences, Franklin Lakes, NJ, USA) and data analysis was performed using FlowJo vX.0.7 (Tree Star, Ashland, OR, USA).

### 2.8. ADCC Assay

For evaluation of antibody-dependent cellular cytotoxicity (ADCC), infected primary CD4+ T cells were stained with viability (AquaVivid; Thermo Fisher Scientific, Waltham, MA, USA) and cellular (cell proliferation dye eFluor670; Thermo Fisher Scientific, Waltham, MA, USA) markers and used as target cells. Overnight rested autologous PBMCs were stained with another cellular marker (cell proliferation dye eFluor450; Thermo Fisher Scientific, Waltham, MA, USA). The primary CD4+ T cells and PBMCs were then incubated 20 min before being washed twice in complete RPMI (Gibco). Target cells (T) were then mixed with PBMC effector cells (E) at an effector/target (E/T) ratio of 10:1 in 96-well V-bottom plates (Corning, Corning, NY, USA). Guinea pig sera (1:1000) with monoclonal antibodies (17b at 5 µg/mL) and/or 50 µM of BNM-III-170, or equivalent amounts of DMSO, when specified, were added to the appropriate wells. The plates were subsequently centrifuged for 1 min at 300 g, and incubated at 37 °C, 5% CO_2_ for 5 to 6 h before being fixed in a 2% PBS-formaldehyde solution. ADCC was calculated as previously reported [35], using the formula: [(% of GFP+ cells in Targets plus Effectors)-(% of GFP+ cells in Targets plus Effectors plus sera)]/(% of GFP+ cells in Targets) x 100 by gating on infected lived target cells. All samples were acquired on an LSRII cytometer (BD Biosciences, Franklin Lakes, NJ, USA) and data analysis was performed using FlowJo vX.0.7 (Tree Star, Ashland, OR, USA).

### 2.9. Statistical Analysis

Statistics were analyzed using GraphPad Prism version 6.0.2 (GraphPad, San Diego, CA, USA). Every data set was tested for statistical normality, and the information was used to apply the appropriate (parametric or nonparametric) statistical test. *p* values of <0.05 were considered significant; significance values are indicated as * *p* < 0.05; ** *p* < 0.01, *** *p* < 0.001; **** *p* < 0.0001.

## 3. Results and Discussion

### 3.1. CD4-Bound Stabilized Immunogens Elicit Anti-Cluster A Antibodies in Guinea Pigs

In order to elicit ADCC-mediating antibodies targeting conserved regions of HIV-1 Env gp120 subunit, we decided to immunize guinea pigs with the core of the gp120 (gp120_core_), the core of the gp120 with the W69A mutation (gp120_core_ W69A) or phosphate-buffer saline (PBS) as a negative control. The gp120_core_ used in this study was generated by removing the variable regions 1, 2, 3, and 5 [31,32], which results in a gp120 stabilized in the CD4-bound conformation [31]. The W69 residue of the gp120 is a highly conserved residue located at the interface between the inner domain topological layer 1 and layer 2 of the gp120. This residue was found to be a key residue governing Env conformational changes and trimer stability [36,37]. Anti-cluster A antibodies, a class of anti-Env antibodies displaying broad and potent ADCC [34,38,39,40], were shown to recognize two distinct inner domain structure: the layer 1-layer 2 interface [33,34,37,41,42,43] and/or the eight-stranded β-sandwich with the N-terminus [43,44]. Alteration of residue W69 was previously shown to dramatically decrease anti-cluster A antibodies binding, therefore abrogating their ability to mediate ADCC and greatly reducing the ADCC responses mediated by sera from HIV-1-infected individuals [34,37,40]. Of note, this residue has also been shown to modulate anti-CoRBS Ab interaction [34,37,40]. 

Two groups of 6 guinea pigs per group were immunized 4 times, each with the different gp120_core_ immunogens (Figure 1). Group three served as control and was given PBS at the immunization points. To evaluate whether the immunization protocol elicited anti-gp120 antibodies, we evaluated the capacity of antibodies present in sera from immunized guinea pigs to bind to the gp120_core_ by ELISA. No significant differences were observed between the two gp120_core_-immunized groups (Figure 2A), indicating that similar levels of anti-gp120 antibodies were elicited in the two gp120_core_-immunized groups. This response was shown to be specific and significantly higher compared to sera obtained from the placebo group, immunized with vehicle only (PBS) (Figure 2A). To evaluate whether the immunogens elicited anti-cluster A antibodies, we coated the stabilized inner domain (ID2) that can only be recognized by anti-cluster A antibodies [33]. As expected, gp120_core_ immunized animals elicited ID2 reactive antibodies, indicating the presence of anti-cluster A antibodies. Surprisingly, despite the key role of W69 in anti-cluster A antibody interaction, sera from gp120_core_W69A immunized animals also recognized ID2. While we observed a small decrease in ID2 binding, the difference was not statistically significant. Variable regions V1/V2 and V3 were previously shown to stabilize the gp120 unliganded conformation since their deletion stabilized the CD4-bound conformation [31]. Therefore, it is possible that deletion of the V1/V2 and V3 regions, required to generate the gp120_core_ could have compensated to some extent for the W69 mutation. 

To evaluate whether the elicited antibodies were able to recognize the trimeric Env, we transfected HEK293T cells with two plasmids coding for HIV-1 Env_YU2_ ΔCT and human CD4, thereby allowing Env to sample an “open” CD4-bound conformation [34]. We then stained these cells with the sera from the different groups of immunized guinea pigs. Figure 3A shows the representative raw data of 1 serum from the first group which binds more to the Env than the other 2 groups, but only when the Env is in an “open CD4-bound” conformation. As shown in Figure 3B, sera from the two gp120_core_-immunized groups were only able to bind to HIV-1 trimeric Env when it was co-expressed with receptor CD4, meaning that the elicited antibodies only recognized CD4-induced epitopes. Significant differences were observed between the two gp120_core_-immunized groups in their ability to recognize CD4-bound Env trimers, revealing that the first group (gp120_core_) of guinea pigs had a higher amount of CD4i antibodies than the second group (gp120_core_ W69A). Therefore, the presence of the W69A mutation did decrease the overall amount of CD4i antibodies elicited by the gp120_core_ immunogen.

### 3.2. Immunizations with a gp120_core_ Elicit Cluster A Specific Responses that Afford Elimination of HIV-1 Infected Cells by ADCC in the Presence of CD4mc and a CoRBS Antibody

Previously, it was shown that CD4i antibodies present in the sera of infected individuals were able to mediate ADCC against HIV-1 infected cells when Env adopts an “open” CD4-bound conformation [20]. Furthermore, it was shown that two classes of CD4i antibodies (anti-CoRBS such as 17b and anti-cluster A such as A32 antibodies) could stabilize a new Env conformation sensitive to ADCC (in combination with CD4mc) [26]. To evaluate whether the elicited antibodies could bind the A32 (anti-cluster A) or the 17b (anti-CoRBS) epitopes, we evaluated whether these antibodies could block recognition of the “open” CD4-bound Env at the surface of transfected cells by the elicited antibodies. When pre-incubating with A32, a significant diminution in binding of both gp120_core_-immunized groups was observed (Figure 3C), indicating a presence of A32-like antibodies although more antibodies binding the A32 epitopes were observed in the first group (Figure 3E). When pre-incubating with 17b, a diminution was observed with all animals from the first and second group except one from the second group (Figure 3D), indicating a presence of 17b-like antibodies in most immunized animals (Figure 3E). Altogether, these results indicate that more A32-like antibodies able to bind the trimeric Env were elicited by the gp120_core_ compared to the gp120_core_W69A.

In order to assess if the sera from the guinea pigs were able to bind to infected cells and mediate ADCC, we infected primary CD4+ T cells with a previously reported wild-type (WT) HIV-1 strain that encodes all accessory proteins, the R5-tropic (ADA) envelope as well as the *gfp* reporter gene (NL4.3 ADA GFP wildtype (WT)) [34,45]. Since the elicited antibodies in both gp120_core_-immunized groups recognized the “open” CD4-bound conformation, we first evaluated the ability of these sera to bind to infected cells in the presence of the small molecule CD4mc (Figure 4A). Upon addition of CD4mc (BNM-III-170), none of the groups displayed a significant increase in binding, suggesting that the addition of CD4mc was not sufficient to expose the epitopes recognized by the elicited guinea pig antibodies. This could be due to low levels of elicited anti-CoRBS antibodies (Figure 3D–E) by the gp120_core_ immunogens used to immunize the animals. This is consistent with previous observations indicating that the variable regions V1/V2 (missing from gp120_core_) and V3 base are important for CoRBS interaction [46,47]. Considering that this class of antibody is required to synergize with CD4mc in order to expose the gp120 inner domain [29], we tested whether the addition of an anti-CoRBS antibody (17b) could increase the binding of the immunized guinea pig sera. Upon the addition of BNM-III-170 and 17b, a significant increase in binding was observed for both groups but this increase was higher in the gp120_core_ group compared with the gp120_core_ W69A group. This observation strengthens the results obtained in Figure 2 and Figure 3, suggesting that most elicited antibodies are anti-cluster A antibodies, recognizing the inner domain of gp120. In agreement with their capacity to efficiently mediate ADCC against cells exposing Env in the “open” conformation, we found that sera from gp120_core_ immunized animals were superior to mediate ADCC in the presence of CD4mc and/or 17b (Figure 4B). Altogether, the elicitation of both anti-cluster A and anti-CoRBS antibodies are required to mediate ADCC against HIV-1-infected cells (in the presence of CD4mc). 

In conclusion, gp120_core_ as immunogens fail to induce specific CD4i responses to both clusters A and CoRBS in guinea pig at the levels that in combination with CD4mc could induce effective killing of HIV-infected cells by ADCC. While gp120_core_ immunizations afford induction of cluster A antibodies at the functional ADCC titer, the CoRBS specific response is low and insufficient to act synergistically with cluster A antibodies and CD4mc to mediate ADCC. Augmentation of sera with extra doses of CoRBS-specific antibody 17b is required, in the presence of CD4mc, to induce effective recognition and killing of HIV-infected by cluster A antibodies present in immunized sera. Altogether, our data indicate that strategies aimed at eliminating HIV-1-infected cells in the presence of CD4mc must take into account the presence/elicitation of these two families of CD4i antibodies.

## Figures and Tables

**Figure 1 microorganisms-08-00710-f001:**
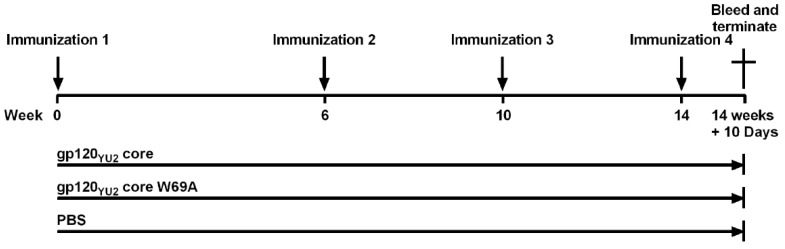
Guinea pig immunization protocol. Groups of six guinea pigs were immunized with each immunogen. A 25-μg sample of purified gp120_core_ or gp120_core_ W69A (or equivalent volume of phosphate-buffered saline (PBS)) was emulsified with AS02A adjuvant in a final volume of 0.5 mL and used to inoculate each animal intramuscularly. Supplemental immunizations were given 6, 10, and 14 weeks after the initial immunization. Blood samples were collected from each animal 10 days after the last immunization.

**Figure 2 microorganisms-08-00710-f002:**
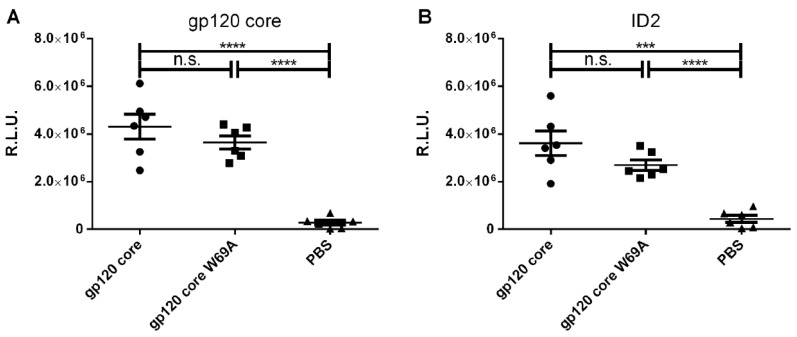
Elicited antibodies recognize the inner domain of gp120. Indirect ELISA on (**A**) gp120_core_ and (**B**) ID2 using immunized guinea pig sera (1:10 000 dilution). Data shown is the mean of four independent experiments. Error bars represent the standard error of the mean. Differences between groups were compared using unpaired t-test (n.s., not significant; *** *p* < 0.001; **** *p* < 0.0001). R.L.U., Relative light units.

**Figure 3 microorganisms-08-00710-f003:**
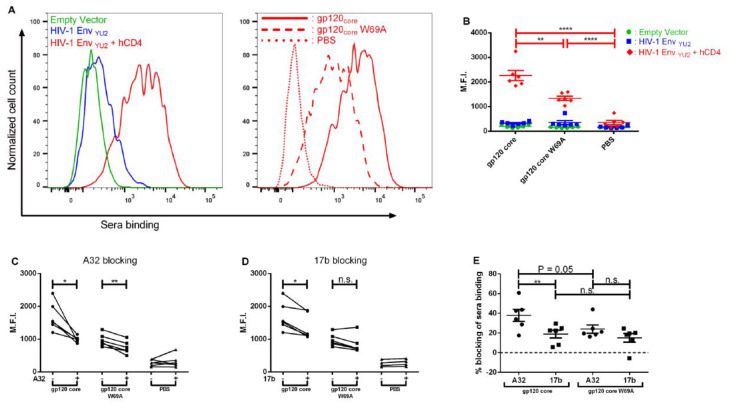
Elicited anti-gp120 antibodies recognize the “open” CD4-bound Env conformation. Cell-surface staining of HEK293T cells either mock-transfected or transfected with an HIV-1_YU2_ Env expressor in combination with a CD4 expressor or not. (**A**) Shown are histograms depicting representative staining obtained with 1 gp120_core_ immunized animal serum on HEK293T cells transfected with an empty vector and an HIV-1_YU2_ Env expressor in combination or not with a CD4 expressor and staining with a representative serum of each group on HEK293T transfected cells with an HIV-1_YU2_ Env and CD4 expressor. (**B**) Cells were stained with immunized guinea pig sera (1:500 dilution) followed by anti-guinea pig secondary antibody staining. Alternatively, cells co-expressing Env and CD4 (i.e., expressing Env in its “open” CD4-bound conformation) were preincubated with A32 (**C**) or 17b (**D**) (5 µg/mL) prior guinea pig sera binding assessment. (**E**) The percentage blocking of guinea pig sera binding with A32 or 17b was determined by calculating ((MFI without preincubation - MFI with preincubation of A32 or 17b)/ MFI without preincubation) x 100. Results are the mean of at least 3 independent experiments. Error bars represent the standard error of the mean. Differences between groups were compared using ANOVA (**B**), paired t-test (**C**,**D**) and multiple t-test (**E**). (n.s., not significant; * *p* ≤ 0.05; ** *p* ≤ 0.01; **** *p* ≤ 0.0001). (−): Without preincubation. (+): With preincubation. M.F.I.: Mean fluorescence intensity.

**Figure 4 microorganisms-08-00710-f004:**
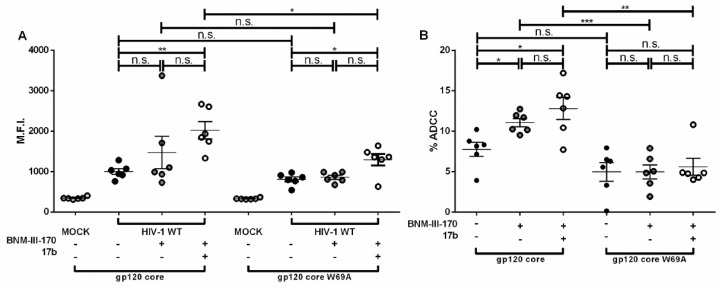
Anti-Cluster A and anti-CoRBS Abs are required to mediate ADCC in the presence of the CD4mc BNM-III-170. (**A**) Staining of primary CD4+ T cells infected with NL4.3 ADA GFP WT using immunized guinea pig sera (1:1000 dilution). (**B**) ADCC quantification against primary CD4+ T cells infected with NL4.3 ADA GFP WT using autologous PBMCs in presence of immunized guinea pig sera (1:1000 dilution). Results are the mean of at least three independent experiments. Bars represent the standard error of the mean. Differences within groups were compared using paired t-test or Wilcoxon test depending on normality distribution. Differences between groups were compared using Mann-Whitney U test. (n.s., not significant; * *p* ≤ 0.05; ** *p* ≤ 0.01; *** *p* ≤ 0.001). (−): Without preincubation. (+): With preincubation. M.F.I.: Mean fluorescence intensity.

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
