# Peer review of "Elicitation of Cluster A and Co-Receptor Binding Site Antibodies Are Required to Eliminate HIV-1 Infected Cells"

_microorganisms, 2020, doi:10.3390/microorganisms8050710_

Round 1

Reviewer 1 Report

In this manuscript Beaudoin-Bussières et., al described the induction of CD4i CoRBS and cluster A antibodies by immunizing gp120core antigen. The authors confirmed the induction not only by comparing binding activities to gp120core and gp120coreW69A but also with the series of binding-inhibition assay using representative mAb to CD4i CoRBS and cluster A. Then the authors tested the plasma samples in binding assay and ADCC assay in the presence or absence of CD4mc. The enhancement in binding and ADCC by CD4mc was only observed when additional CD4i antibody was supplemented in the culture. The results suggested while gp120core immunizations afford induction of cluster A antibodies at the functional ADCC titer, the CoRBS specific response is low and insufficient to act synergistically with cluster A antibodies and CD4mc to mediate ADCC. The experiment was well planned and conducted. The obtained results supports the conclusion that is important for HIV vaccine development.

I have only two minor comments:

Minor points:

  1. Previous reports suggested involvement of variable region such as V1/V2 and V3 base for the binding of antibodies to CD4i CoRBS (Kaplan et al., J Virol. 90: 4481–93, 2016. Tanaka et al., Retrovirology 14:44, 2017). It is necessary to discuss whether gp120core was appropriate for induction of antibodies to CD4i CoRBS.
  2. Line 301; (Figure B) should be (Figure 4B)

Reviewer 2 Report

The manuscript „Elicitation of cluster A and co-receptor binding site antibodies are required to eliminate HIV-1-infected cells” by Beaudoin-Bussières and co-authors aim at characterizing the relative contribution of different classes of non-neutralizing CD4-induced antibodies (anti-CoRBS and anti-cluster A), for antibody-mediated recognition of HIV-1 Env and elimination of HIV-1-infected cells. To explore this aspect, the authors generated sera by immunizing guinea pigs with HIV- gp120 cores, either wild-type or containing the W69A. W69A mutation was previously shown to decrease anti-cluster A antibodies binding to HIV-1 Env. However, the impact of this mutation on binding to anti-CoRBS antibodies seems to not have been shown.

Sera from both immunized animal groups displayed a similar ability to bind to soluble gp120 core in an ELISA, in contrast to serum from PBS-treated guinea pigs. Also, both sera bound to coated ID2 that is recognized by anti-cluster A antibodies. To test the ability of the sera to bind trimeric Env, HIV-1 Env was expressed in cells with CD4, resulting in expression of Env in an open, CD4-bound state. Indeed, CD4 expression was required to make Env accessible to antibodies in both sera. Strikingly, the ability of gp120 (W69A) sera to bind to Env was reduced compared to that of gp120 wild-type sera, indicating that the W69A mutation resulted in elicitation of less antibodies recognizing the open gp120 or in elicitation of antibodies of lower affinity to the open gp120. The authors do not seem to address which of the two possibilities is true. Preincubation of cells with anti-cluster A antibodies (A32) and anti-CoRBS antibody (17b) diminished gp120 staining by the wild-type gp120 core serum. This procedure also reduced staining by the gp120 (W69) core serum, although to a lesser extent (A32) or in an non-significant manner (17b). These results indicate that A32-like and, to a lesser extent, 17b-like antibodies are elicited upon gp120 core immunization.

Finally, sera failed to bind to HIV-1-infected CD4+ T-cells both in the presence and in the absence of the small molecule CD4mc. However, addition of anti-CoRBS antibodies (17b) increased binding of sera. Increase was more pronounced in the gp120 core wild-type group compared to gp120 core (W69A). This result is in line with the idea that sera contain insufficient levels of anti-CoRBS antibodies, and that they mostly contain anti-cluster A antibodies). Sera from the gp120 core wild-type group induced ADCC when CD4ms and anti-CoRBS antibodies were added.

In general, the paper presents interesting data, but lacks a clear flow about which hypothesis is to be tested, and the wording could be improved a bit for clarity and better understanding. A discussion is lacking. The reviewer is convinced that addition of some control experiments and representative raw data would improve the quality of the overall study.

Major points:

  1. A quantification of total amount of IgG in the different sera (including gp120-nonspecific IgG) would be very informative.
  2. There seems to be a conceptual problem since it is stated that “alteration of residue W69 was shown to dramatically decrease anti-cluster A antibodies binding”, however this does not seem to be supported by data shown in Fig. 2B. Please clarify.
  3. In Figure 4, negative controls need to be included in order to show the background of the assays (FACS staining of uninfected CD4+ T-cells assessed in parallel; ADCC assay with uninfected cells performed in parallel to the one with infected cells)
  4. Figure 4: it would have been interesting to add, as separate condition, and in combination with CD4ms +/- 17b, the A32 antibody. This would indicate whether the amounts of elicited anti-cluster A antibodies can be boosted by additional ones.
  5. A discussion must be added. More background information on W69A in the context of antibody elicitation and ADCC is required. Also, there is no clear effort yet visible to discuss the findings in context to antibody responses in HIV-1-infected individuals.

Minor points:

  1. Typo in line 232: “recognized” instead of “recognize”
  2. The reviewer believes that “facilitates the elicitation of anti-cluster A antibodies” (line 235) should be exchanged with “reduces the elicitation of anti-cluster A antibodies”.
  3. Adding representative dot plots or histograms of the FACS-based Env staining and ADCC assay are required to demonstrate the accurateness of the quantifications
  4. Fig. 4. The title of the figure implies that a conclusion on the necessity of having anti-cluster A antibodies in the mix is shown. Is this an indirect conclusion through the comparison with the W69A condition, which has been previously shown to reduce elicitation of anti-cluster A antibodies? However, this idea is not supported by the data shown in Fig. 2B, since there is no significant difference of ID2 binding by the two sera. Please clarify.
  5. Fig. 2, it is stated that “data shown is the mean of four independent experiments”. What does each symbol (n=6) stand for? The same kind of question applies to Figs. 3 and 4, where there are more symbols that numbers of experiments stated in the legend.

Round 2

Reviewer 2 Report

I am satisfied with the introduced changes.